# Identifying Key Factors for Accelerating the Transition to Animal-Testing-Free Medical Science through Co-Creative, Interdisciplinary Learning between Students and Teachers

**DOI:** 10.3390/ani12202757

**Published:** 2022-10-13

**Authors:** Fatima Zohra Abarkan, Anna M. A. Wijen, Rebecca M. G. van Eijden, Fréderique Struijs, Phoebe Dennis, Merel Ritskes-Hoitinga, Ingrid Visseren-Hamakers

**Affiliations:** 1Faculty of Science, Radboud University, Radboud Honours Academy, 6525 AJ Nijmegen, The Netherlands; fatimazohra.abarkan@ru.nl (F.Z.A.); frederique.struijs@ru.nl (F.S.); 2Faculty of Medical Science, Radboud University, Radboud Honours Academy, 6525 AJ Nijmegen, The Netherlands; anna.wijen@ru.nl; 3Institute for Management Research, Radboud University, Radboud Honours Academy, 6525 AJ Nijmegen, The Netherlands; rebecca.vaneijden@ru.nl (R.M.G.v.E.); phoebe.dennis@ru.nl (P.D.); 4Faculty of Veterinary Medicine, Utrecht University, 3584 CL Utrecht, The Netherlands; 5Department of Clinical Medicine, Aarhus University, 8000 Aarhus C, Denmark; 6Institute for Management Research, Radboud University, 6525 AJ Nijmegen, The Netherlands; ingrid.visseren@ru.nl

**Keywords:** animal research, animal-free innovations, new approach methods, NAM, transition analysis, medical research, animal ethics, translation science

## Abstract

**Simple Summary:**

In 2021, the European Parliament called on the European Commission to create an action plan to phase out animal experiments in the European Union. This call for action came because many animal tests continue to be performed despite the introduction of alternatives and efforts to reduce the number of animals used and refine the way animals are used in medical science. An honours project was organised between May and September 2021 at Radboud University to contribute to the acceleration of the transition to animal-testing-free medical science. Teachers and experts delivered lectures on topics related to animal testing, transitions, governance, and legislation. In addition, the students conducted a desk study (literature review and document analysis) and, on 26 July 2021, held nine focus group sessions, each group consisting of five to six experts within various fields. This article analyses which factors could contribute to accelerating the transition to animal-free medical science. We identified six key areas that could support this acceleration.

**Abstract:**

Even with the introduction of the replacement, reduction, refinement (the three Rs) approach and promising technological developments in animal-testing-free alternatives over the past two decades, a significant number of animal tests are still performed in medical science today. This article analyses which factors could accelerate the transition to animal-free medical science, applying the multi-level perspective (MLP) framework. The analysis was based on qualitative research, including a desk study (literature review and document analysis), lectures from experts, and nine online focus group sessions with experts on 26 July 2021. These were undertaken as part of an honours project between May and September 2021 to identify barriers, levers, and opportunities for accelerating this transition. The MLP framework identifies required changes at three levels: innovations and new practices (niche level), the current (bio)medical research system (regime level), and larger societal forces (landscape level). All three levels interact in a non-linear fashion. The model enabled us to identify many relevant factors influencing the transition to animal-testing-free medical science and enabled priority setting. Our findings supported the formulation of six “focus areas” to which stakeholders could devote efforts in order to accelerate the transition to animal-testing-free medical science: (1) thorough and translatable new approach methods (NAMs) for human-relevant medical research; (2) open science and sharing data; (3) targeted funding for NAMs; (4) implementing and modernising legislation for NAMs; (5) interdisciplinary education on animal-testing-free medical science; and (6) facilitating a shift in societal views, as this would be of benefit to both animals and humans. It is proposed that these focus areas should be implemented in parallel.

## 1. Introduction

On 16 September 2021, the European Parliament called on the European Commission (EC) to create an action plan to phase out animal experiments in the European Union (EU) [1]. This call reflected evolving views towards non-animal research, which are motivated by medical [2,3,4], methodological [4,5], and ethical [6,7,8,9] arguments. Furthermore, this call reinforces the three Rs approach, namely to replace, reduce, and refine the use of animals, which has been the legal obligation in EU legislation since 1986 [10]. Russell and Burch’s (1959) 3Rs approach has been the key strategy used, aiming at the achievement of humane experimental techniques [11]. Despite this ongoing scientific, political, and societal attention towards animal-free approaches, many animal tests continue to be performed.

In order to contribute to accelerating the transition to animal-free medical science, an honours project was conducted at Radboud University from May to September 2021, facilitated by Visseren-Hamakers and Ritskes-Hoitinga. The current article is based on the research conducted during the honours project and has two aims. Firstly, we aimed to explore key factors for accelerating the transition to animal-free medical science. The explored factors were organised into the *barriers* that need to be overcome (i.e., what stands in the way of accelerating the transition); the *levers* that have been used (i.e., what actions have been taken to accelerate the transition); and the *opportunities* which could be used in the future (i.e., what possibilities could arise or be created to accelerate the transition) [12]. Secondly, the article aims to reflect on lessons learned from the interdisciplinary co-creative learning process between students and teachers.

We applied the multi-level perspective (MLP) framework [12], which can be used to enhance our understanding of why transitions materialise in the way they do and thereby distil opportunities to accelerate desired transitions. Transition literature is part of a broader body of literature studying societal transformations, transformative change, and transformative governance. This literature argues that fundamental societal changes are required to address sustainability issues, including climate change, biodiversity, pollution, and environmental justice concerns [13]. Animal concerns are increasingly incorporated into these debates, and this lens of fundamental societal change is also increasingly applied to animal-related issues [14,15]. In essence, this fundamental change means addressing the underlying societal causes of sustainability issues. By applying a transition lens, we focused our analysis on those factors (*barriers*, *levers*, and *opportunities*) that represent such underlying causes.

According to Geels [16,17], transitions are nonlinear processes that are the result of an interplay of developments, divided between three levels:
-The *niche* level gives rise to innovative social, economic, technological, or policy-related practices that mature outside of the ‘normal’ market selection in the regime [15,16,18].-The *regime* level represents prevailing practices. This relates to the technological, cultural, political, scientific, market, and industrial dimensions that perpetuate a particular practice [16,17,18].-The *landscape* level encompasses significant social changes in politics, culture, and world views or natural characteristics that are generally slow to change. Landscape developments result from the ideas and actions of large numbers of players and include the public’s risk perception [16,17,18].


From this transition analysis, we inferred six focus areas, which were identified as the most important and urgent factors for accelerating the transition to animal-testing-free medical science.

## 2. Materials and Methods

The current study was initiated through a Radboud University honours project and encompassed a qualitative research design, including a desk study (literature review and document analysis), lectures from experts, and nine online focus group sessions on 26 July 2021. Documents used in the desk study were retrieved through a literature search in PubMed, Embase, and Web of Science, supplemented by documents from colleagues and lecturers during the honours project.

During the honours project, ten students from disciplines including Biomedical Sciences and Environment and Society Studies and two professors and a PhD candidate with backgrounds in Laboratory Animal Science and Environmental Governance collaboratively decided on the research process and design. The professors provided an overview of the transition’s developments as well as the theory on transitions and transformations and provided contacts for further expert knowledge. After that, the students mostly decided among themselves on how to organise the project, working in groups as well as individually on the project. Five experts within the fields of Environmental Governance, Policy, and Medical Sciences provided lectures on the following topics: (1) animal testing and the three Rs, (2) transitions and social scientific perspectives, (3) animal ethics and governance, (4) legislation and current developments in the European Union, and (5) animal experimentation and its alternatives in the Netherlands. All lectures were delivered between 21 May and 18 June 2021.

Additionally, a tour was organised at the Animal Research Facility of Radboud University for teachers and students, and conversations were held with two researchers who were involved with animal research to understand better the realities of animal research practices.

Furthermore, on 26 July 2021, focus group sessions were held online at Radboud University. The focus group sessions were attended by 15 national and international academic and practitioner experts, all active or interested in the field of animal-free medical science and/or (the transition to) animal-free alternatives. Participants included four scientists (in the field of biomedical research, toxicology, and environmental governance); one representative of an international non-governmental organisation (NGO); one politician; one ethicist; one representative of a research foundation; one manager of a research facility; one representative of a scientific database of animal studies; and one philosopher. The focus group sessions started with an introduction to the MLP framework, after which the sessions were hosted in online breakout groups of five to six participants. The participants were randomly assigned to an online breakout group to promote interaction between different participants and stimulate discussion. The topics of the focus group sessions were based on lectures and input from experts and document analysis. There were three sessions, and in each session one topic was discussed by three focus groups, resulting in nine focus groups in total. The three topics were: (1) ‘how could legislators be involved, or legislation be changed, in the acceleration of the transition to animal-testing-free medical sciences?’; (2) ‘how can open data support the acceleration of the transition to animal-testing-free medical sciences?’; and (3) ‘how could education and public communication be changed to accelerate the transition to animal-testing-free medical sciences?’. Throughout the article the focus groups will be referred to as ‘Radboud Focus Group, 2021′.

## 3. Results and Discussion

Below, we organised our results into the barriers, levers, and opportunities that are relevant to the niche, regime, and landscape levels. Using this approach helped us to identify and address the underlying causes that may hold back the transition to animal-testing-free medical science, as well as to find existing levers and opportunities in order to accelerate this transition.

This paper represents a preliminary analysis that can be used as a basis for further research, which is needed to provide more detailed insights and recommendations for each focus area for the specific uses of animal testing and/or different countries.

### 3.1. The Niche Level

At the niche level, the major barriers, levers, and opportunities include medical advances and the assessment of the validity and translatability of models (Table 1).

#### 3.1.1. Medical Advances

While major advances are being made in the field of developing new approach methods (NAMs) in medical science, two important barriers include the development of NAMs in certain areas of research and a lack of validation and implementation of existing animal-free models [19,20]. This may be due, in part, to both validation and implementation being expensive, laborious, and frequently underfunded processes [21]. Generally, the development of NAMs is seen as an exciting scientific innovation and receives much interest from the scientific community. However, the subsequent validation of NAMs receives less praise, since it is sometimes perceived as a rather dull field within science [21].

In addition, for validation purposes, it is often necessary to compare NAMs with animal data instead of the ‘gold standard’ reference, namely in vivo human data. Because whole-organism animal data are not always readily available, in the current regime, the performance of additional animal testing may be required to illustrate the credibility of NAMs, counteracting the goal of using NAMs [21,22]. This is despite the fact that animal studies have never been validated according to the newly formulated strict requirements for NAMs [23]. Furthermore, in the current regime, the lack of implementation of animal-free models is associated with the collective belief in the translational value of animal studies, which are still seen as the best comparator, despite the fact that there is abundant evidence for their reproducibility and translatability issues, such as the reproducibility crisis [24,25,26].

As any model has its limitations, so do 21st-century in vitro systems. For example, in vitro models still lack the integration and longevity of an intact organism, models are designed to simulate human-specific biology with a certain level of complexity, and therefore each system’s relevance for a certain scientific problem must be carefully considered and verified [27]. However, *levers* at this level also exist. Indeed, early human microdosing trials are already available [28], and more sophisticated multi-organ-on-a-chip models are under development [29], which have the potential to counteract the lack of in vivo data that NAMs are currently facing. In addition, there has been an increase in the number of studies that explicitly mention non-animal methodologies over time. In 1997, only 212 studies mentioned NAMs, while 1219 studies named NAMs in 2017, which demonstrates the considerable growing interest in NAMs [30]. A dermatological study suggested that using multiple public datasets containing human data may enable researchers to test and refine their hypotheses prior to starting animal experiments [30]. A study on the effects of performing systematic reviews (SRs) on the attitudes and study quality of researchers showed an increased awareness of the limitations of animal studies [31]. The interviewed researchers noted the poor quality of animal study reports and learned how to improve study quality. In addition, this study resulted in some researchers deciding to move to clinical trials immediately, instead of performing new animal studies first [31]. Convincing examples, such as machine learning for read-across toxicity testing and in silico drug trials, have already demonstrated a higher predictive value for humans compared with animal studies [32,33].

*Opportunities* to promote further innovation and validation of NAMs lie in the funding of these models, as well as the formulation of a list of clear transition objectives for different fields of research that aim towards phasing out animal testing. An example of such a list has been proposed by the Dutch National Advisory Committee on Animal Testing Policy [34]. It is essential to realise that one-to-one replacement (replacing one animal test with one alternative test) may not be the only way forward and that an entirely new approach should be adopted, for instance by combining in vitro and in silico tests or by reformulating research questions [35,36].

#### 3.1.2. Assessment Validity and Translatability of Models

The current method for assessing the relevance and translatability of NAMs, namely positive predictive values (PPVs), has limitations, as it does not capture all relevant parameters for toxicity screening [37]. The parameters measured by PPVs reflect the sensitivity, i.e., the probability that human toxicity was correctly identified by an animal model compared with animal toxicity. This measure excludes the specificity of models, thus missing false negatives, making it hard to assess how effectively non-toxicity in animal models can predict non-toxicity in humans. A better indicator of the relevance of an animal model that adds evidential weight to toxicity tests would be the likelihood ratios (LRs). LRs capture both the sensitivitiy and specificity of a model [37]; however, using the LR requires both human and animal data. As the same invasive research methods used on animals are often not permitted on humans, there are limits to the use of LRs. In addition, often only positive (pre-clinical trial) data are published, leading to biases when assessing LRs [38]. Therefore, a *barrier* to the transition is a lack of proper assessment methods for NAMs and animal models, and suitable data for performing the necessary statistical analyses are scarce.

There are an increasing number of initiatives working towards more open data, pre-registration, and the use of systematic reviews (SRs), which form a *lever* to accelerate the transition to animal-free testing in science [39,40,41]. These movements may bring about changes at the regime level. Open data increases the amount of freely available data for validity assessment. The pre-registration of studies increases the transparency and quality of research, as the aim and the type of data to be collected by a study are disclosed beforehand. SRs help to assess the quality and validity of published data, provide evidence for the (lack of) translatability of the used models, and can identify more suitable alternatives [42]. Several initiatives for pre-registration and publicly accessible databases exist, such as preclinicaltrials.eu and UK biobanks.

*Opportunities* still lie in educating researchers on how to best conduct SRs, perform pre-registration, answer research questions, and formulate more appropriate research questions while also enforcing certain actions to encourage as much open data as possible, as is later discussed at the regime level. Indeed, focus group attendees noted the inconvenience for individual researchers to set up and finance systems to enhance open data, and the hesitancy to share data publicly by companies due to competitive environments [43].

### 3.2. The Regime Level

At the regime level, we identified barriers, levers, and opportunities related to economics, regulation, communication, and education (Table 2).

#### 3.2.1. Economics and Funding Programs

A *barrier* at the regime level is that currently, more funding is available for experiments using animal models than NAMs [2,20], but the development and validation of NAMs also require significant investments. As a result, increased funding and investments in animal models could serve as lock-in mechanisms, which have been shown to make the abandonment of these current techniques in sustainability transitions difficult [44]. Furthermore, as mentioned above, most of the funding that is contributed to NAMs goes to the development of animal-free models rather than their validation [21].

Nonetheless, in an EU-wide survey conducted in 2020, 70% of adults in the EU Member States thought that the EU should invest more in the development of alternative methods to animal testing [45]. At a European Commission Scientific Conference held on 6–7 December 2017 on non-animal approaches, it was also concluded that there is currently no clear plan regarding knowledge resources for the three Rs, and investment is required specifically for areas where animal-free alternatives are still lacking [20]. A *lever* is that in addition to monetary savings in the experimental phase, alternative models can achieve higher predictive results for humans [2]. A higher success rate in the drug development pipeline can also present economic benefits to pharmaceutical companies that are struggling with declining returns on investments [46,47]. Markets for NAMs are expanding, such as in vitro toxicity testing [48], and as NAMs become more widely available and scalable, they become even less expensive, offering researchers a financial incentive to transition. The fact that the Comirnaty vaccine could be conditionally approved for the market one year after the discovery of SARS-CoV-2 [23] suggests that the more efficient and faster approval of NAMs is possible and reliable when given enough priority and funding. Therefore, an *opportunity* may lie in developing a funding strategy for NAMs that includes the comprehensive validation of models before commercialisation.

#### 3.2.2. Regulatory Requirements for Animal Studies

The regulations for animal toxicity assessments are an aspect of the current regime that creates a *barrier* to the transition to applying NAMs [49]. While there are no hard laws against using NAMs and they are accepted on a case-to-case basis, these models are first required to demonstrate that they have the same predictive value as animal models [21]. This validation process is complicated, time-consuming, and costly [23]. Furthermore, it has been shown that validated alternatives can take a long time to finally be implemented in legislation. For example, it took 25 years before the alternative for the rabbit pyrogen test became included in the European Pharmacopeia [50]. Nonetheless, regulations for animal protection exist, namely the EU Directive 2010/63/EU, which states that animal studies are not allowed to be performed when animal-free alternatives are available [51]. However, during the focus group sessions it was noted that what defines a ‘suitable alternative’ is not clear. Therefore, legislative bodies cannot actually enforce this law, and there is a credibility gap between the current possibilities of NAMs and the methods, mainly animal tests, accepted currently by regulatory bodies [43,52].

The fact that political parties are increasingly promoting animal-free research offers a *lever* in terms of regulation. Political engagement resulted in a 2012 motion in the Dutch Parliament to make systematic reviews the norm for preclinical studies—they had already been the norm for clinical studies since the beginning of the 1990s—and led to a 2014 motion to make SRs a mandatory subject in laboratory animal science courses [49]. As a result, an e-learning module on preclinical SRs was developed, which is now a required component of nearly all Laboratory Animal Science courses in the Netherlands. This resulted in more funding for the advancement of SRs themselves. However, the motion to make SRs the norm never came into effect [52]. In 2018, the initiative “Transition Programme for Innovation without the use of animals” (TPI) was set up in the Netherlands [53], followed by Young TPI in 2022 [54]. This initiative is a collaboration of various parties involved in the transition, with the goal of promoting the development of alternatives, and it is coordinated by the Ministry of Agriculture, Nature and Food Quality [53]. In June 2022, the Dutch Parliament adopted another eight motions aimed at accelerating the transition to animal-free research, as it regarded progress too slow.

An *opportunity* for facilitating the quicker implementation of validated NAMs in legislation is to generate scientific confidence in NAMs for the regulatory assessment of the effects of chemicals on human health, both at the national and international level [55]. To help navigate the complexities of the interpretation of Directive 2010/63/EU, legislators should define what is considered an alternative and clarify when an animal test is considered a last resort, as reported by several focus group participants [43]. In the United States (US), the Environmental Protection Agency (EPA) has committed to reducing animal test requirements by 30% by 2025 and eliminating animal test requirements by 2035 [56]. In addition, a bill has been introduced in Congress to change the Food and Drug Administration (FDA)’s rules so that non-animal, human-relevant methods can be used to investigate the safety and effectiveness of a drug, and for other purposes [57]. This was already allowed on a case-by-case basis, but it will now officially be put into the law along with animal tests. Thus, the modernisation of current regulations that require animal testing for new (chemical, cosmetic, and drug) products offers another *opportunity*.

#### 3.2.3. Communication within the Scientific Community

Another aspect of the current regime is the communication within the scientific community. Most of these *barriers* are rooted in the negative consequences related to the perceived competitive nature of scientists. Although competition in science can enable innovation, it also puts pressure on scientists to publish more papers in higher-impact journals, and thus scientists are more likely to perform more research based on their previous findings [58]. Scientists under pressure who work with animal models are likely to perform more experiments quickly, leaving little time to perform SRs or to research alternative methods. Therefore, the pressure of scientific competition could lead to avoiding the use of animal-free models. In addition, the pressure to publish positive data restricts the publication and communication of non-positive study results, leading to the unnecessary repetition of both animal studies and studies to find alternatives [59].

The possibility of registered reports, a relatively new option being adopted by an increasing number of journals, is an appealing incentive that provides a new *lever* [60]. These reports represent a publishing format that emphasises the importance of the research question and the quality of the methodology by conducting a peer review before data collection [60]. Guidelines on how to perform SRs correctly and pre-registration are of vital importance if we are to move towards a research community with more open data. This will contribute to improving study design and the quality of research, reducing the number of animals used in research, and making it easier to find alternatives [31]. Another *lever* is initiating collaborations with various stakeholders and disciplines to coordinate efforts to develop innovative solutions [61]. The honours project that gave rise to this article is an example of such an effective collaboration. The collaboration between students and teachers across various disciplines enabled members of the project to view the transition to animal-free medical science from multiple stakeholders’ standpoints and perform a thorough transition analysis across all relevant topics, from technical and medical innovations to relevant legislation and societal values. In addition, it enabled the honours project members to develop innovative, transdisciplinary ideas on how to connect different levels of the multi-level perspective (MLP) framework and further create opportunities at each level.

The added value of teaching and collaborating in multi-, inter-, and transdisciplinary settings was recognised by the focus group attendees and by the members of the honours project as an opportunity. With improved interdisciplinary communication and cooperation, knowledge crossovers from different areas and disciplines can be utilised to develop more integrative solutions.

#### 3.2.4. Education of Students in the Biomedical Field

Educational programmes in the biomedical field often incorporate animal testing, regularly communicating this as the ‘gold standard’ to students, as noted by the participants in the honours project. The representation of animal studies in courses and early involvement in such studies “normalises” animal research in higher education, whereas education on working with, developing, and validating existing NAMs is lacking, thus presenting a *barrier* to the transition.

An example of how education can be *leveraged* to accelerate the transition to animal-testing-free inventions is the Center of Alternatives to Animal Testing (CAAT). This centre has established a certificate program in humane sciences at the Johns Hopkins Bloomberg School of Public Health, with programs providing courses on alternatives to animal testing [62]. In addition, the EU has initiated the development of new e-learning modules that are freely available via the ETPLAS (European Training Platform in Laboratory Animal Science) platform, stimulating good design in animal research and education in animal-free research [36,63].

Expanding the range of programs and courses on the research into and development of alternatives, and integrating these into the mainstream curriculum, provides an *opportunity* to change the paradigm of animal models as the gold standard in education. In addition, emphasising the relevance of human data within the biomedical curriculum and teaching more in silico approaches, alongside finding “other” methods to answer research questions or asking different research questions, could also contribute to accelerating the animal-testing-free transition [64].

### 3.3. The Landscape Level

The landscape factors (Table 3) identified include views on human–animal relations, views on health in the medical field, and risk perception.

#### 3.3.1. Human–Animal Relations

Both the desk study and the results from the focus groups showed that the current dominant view is that non-human animals are considered in an instrumental manner. This means that animals are valued only for what they can provide for humans rather than having any intrinsic value themselves, and that animals do not have rights and are not recognised as political agents [8,65,66,67,68]. This view is solidified in law, (economic) practices, culture, and through animal ethics committees. Even though attention is paid to reducing suffering through the three Rs, instrumental animal use itself continues to be legitimised. The focus group participants also mentioned another *barrier*: that the public is often uninformed and unaware of animal testing [43]. Moreover, this transition is driven more by politics than by society [69]. Surprisingly, consumers and patients play only a minor role in the transition. They do not have sufficient knowledge or awareness, because there is no accessible labelling or up-to-date information about the use of animal testing in the development of their medication, offering them limited choice and restricting their agency to make informed decisions [43].

Still, a *lever* can be found in the shift in animal–human relations over recent decades, with our relationship to different animals being seen from a less instrumental perspective [69]. This has led to more frequent public discussions on the use of animals in experiments, with a large majority of EU citizens wanting to end animal experiments [45]. The focus group participants also mentioned that the public should play a greater role in the transition, for example through citizen fora [43].

While political attention is important for successful transitions, societal pressure is essential in order to propel change [69]. *Opportunities* lie in shifting the current anthropocentric perspective [67] towards a more inclusive consideration of animals through an ecocentric or biocentric perspective. This would require a radical shift in thinking, feeling, and acting, which, according to the focus group participants, could be achieved by starting education on human–animal relations at a young age [43]. A promising development is the initiative of the EC Joint Research Centre for teaching programs on the Three Rs and Animal Use in Science together with the European Schoolnet, a network of 34 European Ministries of Education that aims to bring innovation to teaching and learning in high schools, creating awareness on these topics at the high-school level [70].

#### 3.3.2. Curative Health Focus and Market Mechanism in the Medical Field

Until recently, the field of medical sciences has focused mostly on curative health. In practice, this means that researchers in this field are primarily occupied with developing treatments to cure illnesses, in large part using animal testing, rather than preventing them [3], which presents a *barrier*.

Changing views on health can be seen as a *lever* for the transition to animal-testing-free medical science. Examples can be found in the James Lind Alliance in the United Kingdom (UK), where patients are involved in determining research goals, which are often something other than new medication [71]. A shift towards focusing on prevention can also be seen in the Netherlands, with the Dutch Cooperating Health Funds aiming to achieve the healthiest youth in the world by 2040 through a focus on prevention [72].

Initiatives that focus on preventing lifestyle-related conditions could be an *opportunity* to decrease the number of animals used for finding cures for those illnesses. This can be achieved through the provision of more training and education for physicians on nutrition, increasing awareness around the complex and interacting aetiologies of diseases and the acknowledgement that exercise, diet, and psychological treatment, in addition to medicine, have the ability to help those who are unwell [73]. The One Health and One Welfare approaches can also contribute to a more holistic view of health that also includes environmental issues, which are important for the prevention of, e.g., zoonotic diseases [74].

#### 3.3.3. Risk Perception of the Public

The demand for animal experiments in medical science is partly perpetuated by society’s perception of risk and concern with how well the government can safeguard patients and citizens when developing or testing new treatments [75]. In science, the status quo is generally more comfortable, as it ensures a ‘known’ level of public safety: this presents a prominent *barrier*. The public is not aware that their perceived risk is often inconsistent with the actual risk [34].

On the other hand, a *lever* is the emergence of public interest in the freedom of choice of treatment. The public’s demand for the ability to make one’s own decisions regarding social responsibility and for more transparency from experts shows that there are people who oppose the status quo. We have seen this in the emergence of patients who take it upon themselves to obtain certain methods of treatment, relying on their own risk perception and not that of society [34].

An *opportunity* lies within the area of risk communication. This will require the adoption of a radically different approach to changing the risk perception of the public. To achieve this, more transparency around risks and how health risks are safeguarded against is needed. In order to leverage the heightened public attention regarding animal studies in relation to health, it is necessary to implement effective communication methods. Organising citizen fora consisting of conversations with citizens on the topic of public health could be a suitable platform and communication method [76].

### 3.4. Interactions between Different Levels of the MLP Analysis

The above analysis showed that the transition to animal-testing-free medical science has *barriers*, *levers,* and *opportunities* at different levels, which interact with each other. Numerous stakeholders from different domains are involved in the transition, including civil society, science, legislation, and industry. While the niche level shows that many animal-free models have been and are being developed, there is also a lack of validation, implementation, and acceptance of these models. For innovations within the niche level to become mainstream, the dynamics at the regime and landscape levels regarding *barriers*, *levers*, and *opportunities* need to change. Niche innovations are currently not sufficiently adopted by the regime due to inadequate funding, persistent regulatory requirements favouring animal studies for safety testing, and researchers’ views of animal models being the gold standard. The landscape level shows the overarching societal views and systems that can either hold back or stimulate changes in the animal-testing regime.

Due to the complexity of the interests, goals, and values of the different factors involved in and affected by animal testing, the transition to animal-testing-free medical science requires transformative change. Such fundamental societal change cannot be achieved through single initiatives or governance instruments but requires efforts in multiple locations with various actors at all levels of governance, brought together in ‘governance mixes’ simultaneously addressing the societal underlying causes. All actors who share the ambition of achieving the fundamental change that is necessary to move beyond animal testing can contribute to such transformative governance [13,77].

### 3.5. Recommendations

This paper had two aims: (1) to distil the factors that may accelerate the transition to animal-free medical science, and (2) to draw lessons from our experience with interdisciplinary co-creative learning between students and teachers.

First, our research identified six focus areas that are expected to have the greatest impact on accelerating the transition to animal-free medical science.

#### 3.5.1. Focus Area 1: Thorough and Translatable NAMs for Human-Relevant Medical Research

The current regime relies heavily on animal studies for medical research, which often results in poor translation to humans, also contributing to a reproducibility crisis. To change this, systematically re-evaluating the animal models used and studied is suggested. Furthermore, NAMs should be validated using human data rather than data from animal studies, as animal studies have reproducibility and translation issues and have never been subjected to the strict validation requirements of NAMs.

#### 3.5.2. Focus Area 2: Open Science and Sharing Data

Extending open research principles and collaboration among various universities, research institutes, and the healthcare industry as soon as possible will aid in critical analysis and establishing a benchmark before conducting animal experiments. Sharing positive and negative data from animal studies could avoid publication bias, and ‘open science’ could serve as a starting point. This could be stimulated by encouraging pre-registration and registered reports.

#### 3.5.3. Focus Area 3: Targeted Funding for NAMs

Although it requires an initial investment, switching to NAMs could save resources in the long term, since perpetuating animal experiments is expensive in comparison to NAMs. This could also help reduce the high rate of failure in the translation and reproducibility of data. In addition, NAMs can save time, as illustrated by the expedited approval of the Comirnaty Pfizer/BioNTech vaccine within only one year. Therefore, a targeted funding strategy should be developed to award funding for implementing and validating NAMs.

#### 3.5.4. Focus Area 4: Implementing and Modernising Legislation for NAMs

Due to the fact that existing laws and legislation are occasionally seen as multi-interpretable, it is essential to clarify and implement laws more effectively and thoroughly. For example, it is unclear when an animal test is considered a ‘final resort’, as the EU Chemicals Strategy dictates. Furthermore, while it is mandatory to use alternatives whenever possible according to Directive 2010/63/EU, it is not mandatory by law to invest in the validation of alternatives for research. As a result, NAMs are not used in research unless developers put in the effort to validate them. Therefore, establishing a better-defined system for implementing legislation is needed, for example, by creating explanatory guidelines for regulations, ultimately aiming at global harmonisation.

#### 3.5.5. Focus Area 5: Interdisciplinary Education on Animal-Testing-Free Medical Science

More comprehensive and interdisciplinary education on the technical and ethical problems of animal research and the opportunities for animal-testing-free innovations is needed. In our honours project, we noticed that students gained a new understanding of this complex topic from different perspectives by working with students from other disciplines. As a result, they improved their knowledge and interdisciplinary thinking when dealing with complex problems. This also allowed students to comprehensively approach a topic from many relevant standpoints and develop creative solutions combining insights from different disciplines. Therefore, more interdisciplinary courses and transdisciplinary collaboration on this topic could enable co-creative learning.

#### 3.5.6. Focus Area 6: Facilitating a Shift in Societal Views

For the public to be able to participate in dialogues on animal testing, there should be more transparency and education for the general population about standard practices in animal labs and research. To move away from the narrow focus on curative health, risks, and anthropocentric world views that perpetuate animal testing, we should promulgate views on health and welfare that are preventive and integrative instead. This asks for a collaborative and interdisciplinary approach, cutting across the boundaries of animal, human, and environmental health and wellbeing. This could be facilitated by public health campaigns, transparency in animal testing and alternatives, and inter- and transdisciplinary research. However, it might also require a strong demand for this change involving the rejection of the status quo, such as through demonstrations and the widespread disruption of the current system, as shown by transitions in other societal issues.

For achieving the second aim, the interdisciplinary co-creative learning process of the honours lab proved immensely valuable. Because of the interactive discussions and lectures, both students and teachers learned from each other’s disciplines, broadening their perspectives and allowing them to think outside the box. The students gained an increased interest in transitions and transformations from the literature, specifically regarding the transition to animal-free science. Students learned to create their own methods of learning and researching. The honours project resulted in numerous opportunities for new research and development. For instance, two students joined the board of Young TPI. Moreover, the two professors, supported by two of the honours lab students, successfully developed a research proposal on animal-free safety assessment. In addition, another student attended an honours programme at the University of Oxford to study the transition to animal-free inventions in depth.

## 4. Conclusions

In conclusion, this honours project allowed us to provide a preliminary overview of the key factors for accelerating the transition to animal-free medical science. Based on this overview, we identified six focus areas to which stakeholders should devote their efforts in order to accelerate the transition to animal-testing-free medical science. The six focus areas were: (1) thorough and translatable NAMs for human-relevant medical research; (2) open science and data sharing; (3) targeted funding for NAMs; (4) implementing and modernising legislation for NAMs; (5) interdisciplinary education on animal-testing-free medical science; and (6) facilitating dialogue on societal views, as this will benefit animals, the environment, and humans. This project also demonstrated that collaboration between teachers and students in an interdisciplinary setting allows for a more comprehensive understanding and new perspectives on complex topics such as this transition. Although many obstacles exist to achieving animal-testing-free medical science, numerous efforts and actors have already contributed to this transition. Significant progress in accelerating the transition could be made with the opportunities identified in this study.

## Figures and Tables

**Table 1 animals-12-02757-t001:** Barriers, levers, and opportunities at the niche level.

Barriers	Levers	Opportunities
Lack of knowledge to validate NAMs	Continued development of NAMs and use of human data	Clear objectives to phase out animal experiments
Lack of suitable metrics and data	Increasing open databases	Education and enforcement of open data platforms

**Table 2 animals-12-02757-t002:** Barriers, levers, and opportunities at the regime level.

Barriers	Levers	Opportunities
Initial investment in NAMs	Economic opportunities	Funding programmes
Regulatory requirements and multi-interpretable laws	Political engagement	Modernisation of regulation and implementation
Lack of communication within scientific community	Existing collaborations such as (Young) TPI	Inter- and transdisciplinary collaboration and responsibility
Lack of education	Shift in education	Moving away from animal testing as the gold standard in (medical) education

**Table 3 animals-12-02757-t003:** Barriers, levers, and opportunities at the landscape level.

Barriers	Levers	Opportunities
Anthropocentric perspectives on animal–human relations	Changing attitudes and legislation about non-human animals	Education to shift anthropocentric perspective
Focus on curative health	Changing attitudes towards health	Focus on preventative health
High concern for potential risks	Heightened attention to transparency/autonomy	Investment in risk communication

## Data Availability

Data are available on request by contacting Ritskes-Hoitinga (j.ritskes-hoitinga@uu.nl).

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
