# Peer review of "Identifying Key Factors for Accelerating the Transition to Animal-Testing-Free Medical Science through Co-Creative, Interdisciplinary Learning between Students and Teachers"

_animals, 2022, doi:10.3390/ani12202757_

Round 1

Reviewer 1 Report

Although the topic of this article is very interesting, the methods and results are not well done and often overstated. In terms of the methods, conducting a convenience literature review and unstructured focused group of 6 individuals all of who currently support animal-free testing is not going to discover barriers. It does not seem like a thorough review of the literature was conducted and many references are misused to support statements inappropriately. Additionally, the questions asked of the focus groups led to particular answers rather than truly discovering the most relevant barriers. A huge limitation of the study is having such a small group and not including people who are more thoroughly convinced of animal research and understand why they are currently used.

Several parts of the results include statements that are much too strong. Some of the recommendation are certainly good (e.g., Focus are 1 and 3) but their rationale is not complete. Other recommendations I am also less convinced of their value (Focus area 2, 4, 6). 

In order for this paper to be turned into something acceptable the methods would need to be more fully reviewed especially what literature was reviewed. Sentences/references would need to be carefully fixed to ensure peer-reviewed articles support statements. Finally, all results would need to be much more carefully stated as this is what one small group came up with based on targeted questions. Overall it needs to be written as more balanced. Detailed comments below:

Abstract

·       Replace “perpetuated” with a softer word such as continued

·       Is this just about medical science? This makes it seem just about human healthcare

·       “human-relevant” what about research for animals?

·       What about research on cosmetics, products, etc.?

Introduction

·       Replace medical with “scientific”? Medical feels a bit broad

·       Very interesting to learn about MLP and transition literature

·       Who is “Geels?”

·       “animal-testing-free medical science” > there is some who wonder if this can be done responsibily

Materials and Methods

·       I am doubtful that a single tour can give teachers and students a “complete overview of the procedures involved in practicing animal experimentation.” I also think this wording is not quite right as it makes it sound like animal experimentation is conducted for its own right. Suggest rewording “to better understand what is involved in animal research”

·       A key limitation in the focus group is to not include people who are more firmly conducting animal research and not convinced of animal-free methods. These are the key people that need to be changed and convinced.

·       Focus group

o   What type of scientists?

o   What the manager of an animal research facility?

·       The phrase “animal testing-free medical sciences” is still a bit odd to me. It sounds like that’s the phrase used in the research, but can it be explained why this phrase was used/chosen? Some in the field do not like the term “animal testing?” And most seem to be more familiar with “biomedical research” as medical sciences is more commonly used for active medical practitioners (e.g., doctors, nurses).

·       How were the questions developed? They are quite prescriptive and lead people to particular answers rather than identifying/tackling underlying causes.

Results and Discussion

·       I’m not convinced this approach has helped “identify and tackle the underlying causes holding back the acceleration…” The wording is a bit too strong here.

3.1.1

·       I continue to be thrown off by the term “medical science” instead of biomedical research

·       Lack of validation and lack of implementation are separate problems that I would split

·       Are there NAMs that truly mimic complete, whole body organisms? I am not aware of any. I’d say the first barrier is that these simply do not exist which is important.

·       This statement is much too strong “This is due to the fact that the actual validation and implementation of NAMs, after initial development, is an expensive, laborious, and a frequently underfunded process.”

o   Rephase to “This may be due to, in part, both validation and implementation being expensive, laborious, and frequently underfunded process.”

·       “However, the subsequent validation of NAMs receives less praise, since it is perceived as a rather dull field within science” < I disagree with this statement. I know plenty of people in the field who are very excited by validation and praise it immensely. Rather, the general public/media seems less enthused by it.

·       The citation for the above two statements actually makes a key point that one key barrier is the current state of scientific knowledge and that there is not a NAM that gives a full picture that an animal test does

·       Disagree that a “wide range” is already available. Rephase to “many”

·       Citation 28 and 29 is used improperly

3.1.2

·       The topic sentence comes out of nowhere. How commonly is PPVs used in deciding on NAMs? Again, many researchers I know look at both sensitivity and specificity. There are discussions on going on assessing qualification

·       I’m not sure open data, pre-registration, and SRs actually lever transition. They are good practices that may help support some transition, but they also continue to support animal research.

Table 1

·       The barrier “lack of validated NAM” seems a bit circular”

·       How can we make a clear timeline if we don’t have NAMs developed? Also, there wasn’t any evidence that this is something that is an opportunity.

3.2.1

·       What funding is available for NAMS? It also makes sense that more funding is available for animal models as they are already in place and supporting research so take up more of the field.

·       To use reference 42, you need more context. Right now it makes it seem like there’s a paper showing that this reference is about animal research, but its about carbon

·       The general population wanting more funding doesn’t indicate this is a barrier or opportunity. The scientists would be more relevant

·       HSI is not the most reliable source. Overall NAMs likely cost less across the whole pipeline though.

·       The COVID vaccine example is misleading as there has been substantial animal research conducted leading up to it. There were MANY factors that led to fast approval of these vaccines such as having a large test population with high rates of infection and saying that it means that other therapeutics could be developed faster is not quite right.

·       “Therefore, an opportunity lies in developing a funding strategy for NAMs that includes the comprehensive validation of models before commercialisation.” - Are academic institutions going to take on validation then? That would be great and I think this is an opportunity but there needs more lead-up and explanation about how there is more funding for development vs validation.

·       Furthermore, markets for NAMs are expanding, such as for in vitro toxicity testing [47]. < this statement comes out of the blue. It needs more leadin.

3.2.2

·       Most federal agencies already accept NAM data on a case by case data.

·       “As a result, experiments with animals are still being carried out, when they could have been performed without animals [50].”

o   This statement is simply not true and is also misleading. Experiments are generally not widely performed in animals when there is confidence they could be performed without animals. Rather it takes time for the confidence to build.

·       Is there evidence that these SRs have become more common as a result of the motion? And is there evidence that these motions have actually accelerate the transition? Sometimes those in the field wonder if the legislation does actually help

·       The FDA Modernization Act is widely misunderstood. The FDA already was using non-animal methods in submissions. This just officially puts it in law along with animal tests. Animal tests will still be in the language

·       “Thus, the modernisation of the regulation requiring animal testing for new (chemical, cosmetics, and drugs) products offers another opportunity.” < cosmetics are widely already not allowed to be tested in animals.

3.2.3

·       Not enough information about the “negative consequences” and “perceive competitive nature.” I know plenty of scientists that would like to use NAMs but it can be challenging to learn about them, implement them, or have confidence in using them.

·       “In addition, there is a lack of publication and communication of non-positive study results, which leads to unnecessary repetition of animal studies [55].” < this doesn’t seem about accelerating NAMs but rather about reducing animal tests. It also doesn’t seem connected to the previous sentence or paragraph

·       Registered reports and SRs seem like an opportunity to increase better animal research, not necessary NAMs.

·       The collaboration piece seems to be separate from registered reports and SRs.

3.2.4

·       This normalization is accurate right now. We don’t have enough good, validated NAMs to transition away from animal research right now. I think it would be more accurate to state the barrier of not enough education about NAMs when they exist.

·       Is there evidence that CAAT’s education has led to more implementation of NAMS and transition away from animal research

·       ETPLAS has many modules about the 3Rs and good design for animal research. It is inaccurate to say this platform is about transitioning away. There’s also not evidence of how well this is working.

3.3.1

·       That animals are instruments is not something I hear widely in the community. They are respected and valued for their service by many in the community that have a large amount of empathy.

·       How do animal ethics committees view animals as instruments? They support animal welfare nad scientific quality.

·       I’m not sure increased labeling would actually be good.

·       Many in the scientific community would say that the public should not play a bigger role in the transition because they are not well-informed.

3.2.2

·       I’m not sure I totally understand how focusing on prevention will decrease animal research. Prevention is awesome, but people are still going to want medication for when they are sick. There are many diseases and illnesses that can’t just be fixed with  lifestyle cures.

3.3.3

·       Reference 32 does not support the sentences it is cited afterwards. There is not evidence that the publics perceived risk is inconsistent with actual risk. Citations are necessary for this section.

3.5

·       I don’t see evidence in the paper that these focus areas will make the “greatest impact.” They may help, but there’s not good evidence they are the most key from the limited focus group and literature search.

·       Focus area 1 > evidence for poor translation and contributing to reproducibility crisis has not been well established.

·       Focus area 2 > Still unclear how open science and sharing data helps transition to NAMs as the evidence seems to focus on preclinical SRs and improving current animal research

·       Focus area 3 > Again the COVID vaccine example is misused. Funding is always better

·       Focus area 4 > I don’t think laws are going to be the one to establish which NAMS are accurate. It’s unclear what the recommendation is here.

·       Focus area 6> Nearly everyone I talk to would rather NOT use animals in research. I don’t think focusing on public society is going to help drive the shift. If anything, harassment by a small number of the public of animal researchers only makes them more staunch in their views.

Reviewer 2 Report

I think that this is an excellent manuscript, from which all educational institutions should learn, which want to promote the 3Rs in the life sciences.

However, I am quite disappointed that you are not anywhere in your manuscript mentioning the two pioneers of the 3Rs Bill Russell and Rex Burch, who established in 1959 the 3Rs Principle in their book "The Principleas of Humane Experimental Technique" (Methuen, London) in which they classified humane techniques under the headings of replacement, reduction, and refinement--now commonly known as the three Rs.
Although the Russell and Burch and the 3Rs had been forgotten for more than 20 years, due to the renaissance in the 1980ies of the concept of treating experimental animals as humane companions, 3Rs Centers have been established around the world as outlined in the recent report "The Rise of Three Rs Centres and Platforms
in Europe" http://www.centro3r.it/en/rise-three-rs-centres-and-platforms-europe

Based on the Honours project of the Radboud University the authors are focusing on progress made in Europe and on the Netherlands in particular. However, they should also mention that in 2022 there are many 3Rs centers at universities around the world, which are fosusing on accelerating the transition to animal-testing-free medical science.
